Achieving physical examination competence through optimizing hands-on practice cycles: a prospective cohort comparative study of medical students

Zhang Zinan 1 2
Tang Zhenwei 1 3
Wang Fang 4
Yu Jingjia 5
Tang Youzhou 6
Jiang Boyue 1
Gou Yue 1
Lu Ben 7
Tang Anliu tanganliuxy3@csu.edu.cn 2
Tang Xiaohong tangxh007007@163.com 5 8
1 Xiangya Medical School, Central South University , Changsha , China
2 Department of Gastroenterology, The Third Xiangya Hospital of Central South University , Changsha , China
3 Department of Dermatology, Xiangya hospital of Central South University , Changsha , China
4 Department of Endocrinology and Metabolism, The Third Xiangya Hospital of Central South University , Changsha , China
5 Department of Cardiovascular Medicine, The Third Xiangya Hospital of Central South University , Changsha , China
6 Department of Nephrology, The Third Xiangya Hospital of Central South University , Changsha , China
7 Department of Haematology, The Third Xiangya Hospital of Central South University , Changsha , China
8 The Clinical Skills Training Center, The Third Xiangya Hospital of Central South University , Changsha , China
Bauman Eric
Electronic publication date: 2021 Dec 1
Publication date: 2021
Volume: 9
Electronic Location ID: e12544
Received 2021 May 25; Accepted 2021 Nov 4
Copyright: ©2021 Zhang et al.
Copyright year: 2021
Copyright holder: Zhang et al.
License: This is an open access article distributed under the terms of the Creative Commons Attribution License, which permits unrestricted use, distribution, reproduction and adaptation in any medium and for any purpose provided that it is properly attributed. For attribution, the original author(s), title, publication source (PeerJ) and either DOI or URL of the article must be cited.
License URL: https://creativecommons.org/licenses/by/4.0/

Keywords: Physical examination, Learning curve, Chinese medical education, Deliberate practice, Practice cycles, Medical teaching

Funding: Educational Science in Hunan Province XJK19AGD001 Research project of teaching reform in Colleges and universities of Hunan Province HNJG-2020-0071 The youth funding program of “the 12th Five Year Plan” of Education Science in Hunan Province XJK015QGD014 Research Project on Educational and Teaching Reform of Central South University 2019jy189 2020jy170 This study was carried out with support provided by “The 13th Five-Year Plan” of Educational Science in Hunan Province (No. XJK19AGD001), Research project of teaching reform in Colleges and universities of Hunan Province (HNJG-2020-0071), the youth funding program of “the 12th Five Year Plan” of Education Science in Hunan Province (XJK015QGD014), and the Research Project on Educational and Teaching Reform of Central South University (No. 2019jy189, 2020jy170). The funders had no role in study design, data collection and analysis, decision to publish, or preparation of the manuscript.

==============================
Background

Deliberate practice (DP) was proposed for effective clinical skill training, which highlights focused, repetitive practice and feedback as the key points for practice. Although previous studies have investigated the effect of feedback in DP, little is known about the proper repetitive cycles of clinical skills training especially in physical examination (PE) training.

Methods

We drew learning curves and designed a comparative study to find out the optimal number of hands-on practice cycles, an important aspect of DP, in abdominal PE training for medical students. A comparative study was conducted to validate the optimal number of hands-on practice by dividing students into two cohorts including Cohort A (high-frequency hand-on training) and B (low-frequency hand-on training).

Results

The learning curve study of 16 students exhibited a threshold of four repetitive practices when 81.25% students reached the competence score. A total of 74 students’ final exam scores were collected for analysis. Students in Cohort A (4–5 PEs) scored significantly higher than those in Cohort B (≤3 PEs) (84.41 ± 11.78 vs 76.83 ± 17.51] in the final exam (P = 0.030)).

Conclusion

High-frequency practice can improve students’ competence of abdominal PE skill. We recommend four cycles of hands-on practice for each student in a training course like PE training.

Introduction

Clinical skill performance is considered to be a core competency for medical students and is crucial to professionalism in medical practice. Most importantly, it contributes to successful outcomes in patient care (Zhang et al., 2015). For instance, physicians’ underperformance in physical examination (PE), a basic clinical skill and an important part of a physician’s daily activity (Grune, 2016), has become a frequent cause of medical errors that cause nearly 100,000 deaths per year (Institute of Medicine Committee on Quality of Health Care in A, 2000; Verghese et al., 2015). It has been repeatedly observed that medical students and clinical interns should improve their clinical skills (Krautter et al., 2015; Ramani et al., 2010; Xiaoqing, 2015). In our institution, there are more than 100 pre-clinical students who annually start their clinical skills training at their third year. Usually, 10 to 15 students take a three-hour PE training course together. The mean scores of PE final exams are between 70 to 80 out of 100, far from 90, the predefined level of competency determined by our institution experts. In order to find the reasons for such unsatisfactory results, we distributed questionnaires to 10 medical schools. We found that the average number of hands-on practice cycles among the students was 2.69 based on the student questionnaires. 67.65% of the students thought that the practice cycles in class were insufficient or seriously inadequate (Supplemental Information). Practice cycles seem to be the crux of the problem.

Achieving competence in skills usually requires the involvement of proper practice. According to behaviorism, practice serves as an important part of the psychomotor skill education, and can not only improve the learner’s clinical ability, but also enhance the cognitive enrichment, especially memory formation (McGaghie & Harris, 2018). In recent years, deliberate practice (DP) has been recommended for effective skills training, emphasizing focused, repetitive practice and feedback (Motola et al., 2013). The positive effect of DP in improving PE training has been supported by systematic reviews. However, most studies about DP in PE training focused on the effect of feedback in DP (Criley et al., 2008; Houck et al., 2002; Mookherjee et al., 2013; Smith et al., 2006). There were few studies exploring the appropriate number of hands-on practice cycles, which is another key element of DP (Byrne, Pugsley & Hashem, 2008) in PE training that can help medical students reach the best learning outcomes within limited class hours.

To address this issue, we carried out the research with two aims: (1) to find the optimal number of practice cycles in a PE training course; and (2) to validate the threshold (cut-off cycle) defined by the learning curve, as the abdominal PE an example.

Materials & Methods

The ethical approval for this study was provided by the Ethical Committee of The Third Xiangya Hospital of Central South University, China (2018-S406). The written informed consent was obtained from every subject before the enrollment. All selected subjects volunteered to participate in the study.

Standardization of instructors and examiners

Four instructors and assessors involved in this study were carefully selected by the Department of Internal Medicine of the Third Xiangya hospital. They had similar educational and clinical backgrounds (M.D. and senior physician), which ensured their consistency in abdominal PE experience. Before they taught PE class, they received standardized training to ensure the consistency of their teaching method and scoring. After a pre-class training, five student volunteers performed the abdominal PE and each of them was scored independently by all the assessors using the same assessment instrument to help evaluate the consistency of the assessors’ scores (k = 0.908).

Learning curve study

The flow chart of the learning curve study and comparative study is shown in Fig. 1. Angoff method (Shulruf et al., 2016) was conducted to determine the competence score for the students who took part in the learning curve. A total of five experts were enrolled and determined that the competence score was 90 percent. Procedures of the learning curve study are as follows: First, 16 third-year medical students were recruited voluntarily and instructed how to perform abdominal PE. Then, the students performed abdominal PE on their classmates and were assessed by the same assessment instrument for at least five times. A brief instructor-led feedback was given after each assessment. The scores of participants at each turn were collected for analysis.

Figure 1 The flow chart of enrollment process for students who took part in learning curve study and comparative study.

Comparative study

In order to validate the threshold of the learning curve study, a comparative study was conducted. During the 2019 academic year (From March 2019 to July 2019), all inexperienced third-year medical students who started to learn physical examination in Xiangya School of Medicine of Central South University were eligible for inclusion. None of the students had previous experience in PE. Students who did not sign the consent form of the research program were excluded.

Participants were randomly divided into cohort A and B. The instructor-student ratio of each cohort was 1:12 to 1:14. All participants attended the abdominal PE training course. In the class, the instructors first demonstrated and showed every detail of abdominal PE step by step to make sure that students could independently perform the skill later, and then the students practiced abdominal PE on their classmates. During the practice session, instructors provided feedback based on the students’ performance. Instructors would demonstrate the correct procedure if students made any mistake. Each participant in cohort A tried four or five PEs in class (high-frequency hand-on training), while each participant in cohort B did no more than three PEs (low-frequency hand-on training). The fourth cycle was the cut-off cycle of PE training according to the learning curve study.

Seventy-five days later, participants who were still in the program and did not receive the learning curve study were randomly selected to have a final exam on abdominal PE. The scores of the final exam were used as an indicator to evaluate the performance of the students in the two cohorts. Due to the limitation of test time and the number of examiners, some students were randomly selected to participate in the test. Students who didn’t reach the competence score would accept the remediation, retraining and retest later until they reach the competence score.

Assessment instruments

We developed our own assessment tool based on the scale that our institute used for years in abdominal PE evaluation. As recommended by the Ministry of Health of the People’s Republic of China on Diagnostic Syllabus, and the five domains described by Messick Messick & Series (1990), the reliability and validity of the assessment tool are demonstrated as follows: The content was developed by experienced experts based on the diagnostic textbook designated by the Ministry of education of China (Wan, 2018). The response process was ensured by the same training and exam time for students. The internal structure was demonstrated by unifying the training assessors and by measuring the Kendall’s W value of the assessment instruments.

Statistical analysis

The recorded data were analyzed by IBM SPSS version 23.0 (IBM Corporation, Armonk, NY, USA). The cartogram was drawn by Origin Pro 2019 version 9.6.0.172 (OriginLab Corporation, Northampton, MA, USA). The Kendall’s Coefficient of Concordance was conducted based on the scores of the five student volunteers’ PE performance assessed by all the assessors. The students’ t tests were used for quantitative variables. Chi-square test was used to detect the difference in the attainment rate of competence score in each repetition in the learning curve study. Statistical significance was set at a P value of < 0.05.

Results

Learning curve

In the learning curve study, 81.25% of the students reached the competence score when they performed the procedure at the fourth cycle of practice. Although more students could achieve competence score after practicing five times during the classes, the effect of the improvement of scores at the fifth cycle compared with that at the fourth cycle was not obvious (P > 0.05. Figure 2). Therefore, the fourth cycle was considered to be the cut-off cycle of abdominal PE teaching and was used for the following comparative study.

Figure 2 The relationship between the practice cycles and the percentage of students who reached the competence score.

The X-axis represents the number of cycles, and the Y-axis represents the proportion of students who reached the competence score to all enrolled learning curve study students. (* P < 0.05 when compared to the previous cycle).

Comparative study

A total of 118 students were enrolled in the comparative study, including 59 students in cohort A and 59 students in cohort B. Of the 118 students, 25 students withdrew from the training course and 74 students were randomly selected to participate in the test. Of the 74 students whose data was included in the final exam, 34 (45.90%) were from Cohort A and received high-frequency hands-on training while 40 (54.10%) were from Cohort B and received low-frequency hands-on training. The former scored significantly higher than the latter in the final exam (84.41 ± 11.78 vs 76.83 ± 17.51; P = 0.030; Fig. 3).

Figure 3 Results of the comparative study of low-frequency hands-on training and high frequency hands-on training.

The Y-axis represents the scores of the students’ final exam. Students in high-frequency cohort scored higher than that of low-frequency cohort (84.41 ± 11.78 vs 76.83 ± 17.51; P = 0.030).

Discussion

In this study, we focused on the cycles of hands-on practice, an important aspect of DP, in abdominal PE training through learning curve drawing and validated comparative study. Our results demonstrated that the number of students’ encounters with hands-on practice would influence the outcome of DP in abdominal PE training, and a threshold of four repetitive practices should be recommended for DP designing in abdominal PE training.

It is well known that DP can be used in medical education, especially in simulation-based medical education. A meta-analysis proved that DP in simulation-based courses could improve trainees’ skill acquisition more efficiently compared to traditional training methods (McGaghie et al., 2011). Generally, the core of DP locates at practice and feedback. In this method of training, the learner repetitively practices procedures and undergoes assessment with feedback, resulting in observed improvement of skill performance (Duvivier et al., 2011; Ericsson, 2004; McGaghie et al., 2011). In the long run, a constant skill improvement and fair maintenance can also be observed following this method (McGaghie et al., 2011). Therefore, DP seems to be an ideal means of PE training (Mookherjee et al., 2013). Previous studies focused on the application of repetitive hands-on practice and DP in cardiac auscultation skills, advanced cardiac life support and temporary hemodialysis catheter insertion (Barsuk et al., 2009; Butter et al., 2010; Wayne et al., 2005). However, as for pre-clinical PE training courses, little information could be provided from previous studies. In contrast, our study has explored the potential of repetitive hands-on practice in PE course, a basic skill training applying DP theory. Still, more detailed information should be investigated with practical application of DP in medical training. For instance, Davidson et al. proved a beneficial role of providing first-year cardiology fellows up to 1 h of DP on a right heart catheterization simulator. However, the author also declaimed that the design of DP was not ‘extensive’ and no specific parameters such as numbers of cycle were provided to describe the quality of DP (Davidson et al., 2021). Similarly, Barsuk et al. demonstrated the beneficial role of 2-hour DP designing in central venous catheter (CVC) simulator courses, but no specific information about the numbers of repetition was involved (Barsuk et al., 2009). In this case, we systematically investigate this component of DP designing and we suspected that future studies related to DP should take the numbers of cycle into consideration.

PE training program with DP is to make most students achieve the competence scores required by the curriculum objectives. Although it is widely accepted that practice makes perfect, the training time for every student is limited in reality and the students cannot practice over and over without limitation. Therefore, it is important to determine an optimal number of practice cycles in a time-limited PE class. Learning curve, one of the best ways to evaluate how quickly it takes to acquire new skills or knowledge, has been widely used in researches regarding medical skill learning such as surgical operation (Kockerling, 2018; Lodhi, 2009), patient condition assessment (Cheung et al., 2014), and basic clinical skill (Jiang et al., 2011), et al, but rarely reported in PE study. In general, these studies all showed that a learning plateau would appear after successive and repetitive practice. We assumed that describing the cut-off between the rise and the plateau of learning curve could help optimize course design to balance training efficiency and training time. In this study, we drew a learning curve of DP based PE courses and found that most students (>80%) could reach the competence score after performing four PEs. As a result, we assumed that the optimum number of hands-on practice cycles in a PE training class was four. Our comparative study further validated the assumption, in which two cohorts were established according to the cut-off value. Students from the high-frequency hands-on training cohort practiced four or five times and acquired better scores than their counterparts. The impact of DP on learning was largely determined by the quality of practice. We hypothesized that, in the higher frequency group, students were assigned to practice four or five times, which set a quantitative requirement of DP, and this further facilitated the impact via repetitive practice of DP. Also, since individual feedback was involved in DP, students from high-frequency groups might receive more feedback from instructors, which therefore may help them acquire better competence. Future attention can be paid on detailed requirements of the feedback in DP design.

We believe that the repetitive cycle of DP plays a significant role in the learning efficiency. For better outcomes, the curriculum designers could increase the instructor/student ratio and enlarge the practice portion in a time-limited PE training to ensure that every student can get at least four opportunities to perform PE on standard patients or their classmates.

The study has several limitations: 1. We did not conduct baseline test because participants who had any previous PE training or exposure were excluded at the beginning of the research project. As a result, we assumed that all participants had no experience in PE and their different knowledge or mastery of PE was not considered as a variable. 2. Due to limited time in the practice session and the shortage of teaching resources, we did not conduct a cross-design to ensure that every participant in the project could receive both high-frequency hands-on training and low-frequency hands-on training. Given limitations mentioned above, our team plan to further our study with lager student population that focusing more on the relationship between the numbers of cycle and feedback. Besides, we are interested in examining whether the PE of other regions systems would follow the same pattern.

Conclusions

The number of practice cycles for students in abdominal PE training course should be stipulated to ensure that students have adequate performance chances in class. High-frequency hands-on training strategy can enable students to acquire and maintain PE skill better and help improve their final exam scores. We recommend four hands-on practices for each student in a time-limited PE training course.

Supplemental Information

Supplemental Information 1 Raw data

Click here for additional data file.

Supplemental Information 2 Questionnaire data

Click here for additional data file.

We extend sincere gratitude to all the students and volunteers who have participated in our research program.

Additional Information and Declarations

Competing Interests

Author Contributions

Human Ethics

Data Availability

The authors declare there are no competing interests.

Zinan Zhang, Zhenwei Tang and Anliu Tang conceived and designed the experiments, performed the experiments, analyzed the data, prepared figures and/or tables, authored or reviewed drafts of the paper, and approved the final draft.

Fang Wang, Jingjia Yu, Youzhou Tang, Boyue Jiang and Yue Gou performed the experiments, authored or reviewed drafts of the paper, and approved the final draft.

Ben Lu and Xiaohong Tang conceived and designed the experiments, analyzed the data, authored or reviewed drafts of the paper, fundingadminstration supervision, and approved the final draft.

The following information was supplied relating to ethical approvals (i.e., approving body and any reference numbers):

The ethical approval for this study was provided by the Ethical Committee of The Third Xiangya Hospital of Central South University, China (2018-S406). Informed consent was obtained from every subject before the enrollment. All selected subjects volunteered to participate in the study.

The following information was supplied regarding data availability:

The raw measurements are available in the Supplementary Files.

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
