# Peer review of "Achieving physical examination competence through optimizing hands-on practice cycles: a prospective cohort comparative study of medical students"

_PeerJ, doi:10.7717/peerj.12544_

## Round 0.1 · original submission · Major Revisions

Thank you for your patience as it relates to the reviewing process. Should you choose to revise, please do pay careful attention to the recommendations of reviewers one and three, particularly as it relates to organizational structure and use of tables, figures, etc.

Reviewer 1 ·

Basic reporting

The theme of the reported study is focused on a specific topic (standardized physical examination training) of the medical undegraduate students, which I believe is within the scope of PeerJ publication. The manuscript was generally easy to read and is targeted for the international audience. The authors have provided informative data on regarding the research

However, I think there is still room for improvement regarding the use of terms in area of medical education. And some description in the Methods section needs further clarification.

I feel that contents of Figure 1 can be illustrated in words, therefore Figure 1 seems unnecessary. On the other hand, it would be nice if the authors could include a figure that actually demonstrate the learning curve.

Line 65, "behavioral learning theory" should be "behaviorism".
Lines 125-127, the description of "groups" vs "cohorts" is quite confusing, the authors might need to rephrase this part to allow easier undertstanding for the readers.

Mis-spelling in Figure 2, "Dyas" should be "Days".

Experimental design

The general research focus of the study is on the repetition of the DP. The authors did not seem to present strong rationale regarding the relationship the questionaire part towards the rest of the study, therfore I feel that it's not necessary to include the first part (to determine the reasons why the students cannot master PE skills well in a training course , we conducted a questionnaire survey to the PE instructors and students on their views on the PE courses and the effects of teaching/learning) in the study. Further explaination might be needed as why the survey should be included in the study.

The authors did not provide the reason for why 60 days and 15 days were determined to conduct follow-up assessment and the final exam on abdominal PE respectively in lines 136 and 139.

Validity of the findings

The authors compared their study results regarding repetition cycles of PE towards other procedural trainings in lines 243-248, I think there is a degree of consistency among different studys, therefore, the authors should take the opportunity to expand ond descripbing such consistency and search for further support with learning theories and/or principles. I believe this is the holy grail of this study.

And I feel that the authors failed to explain why "high-frequency hands-on training" is effective or efficient.

Also, I would carefully consider how the result of this study can by broadly applied in other areas of the world (lines 254-255).

·

Basic reporting

Is there any clear definition made by the PE experts about what "mastery" is when it comes to PE conducted by medical students?

Experimental design

It is not clear to me how the students were distributed into 10 groups (line 125) to assure randomisation.
The 60 days following the practice sessions, were students able to practice PE (line 136)? and how were these 25 students recruited?
What randomisation method was used to pick up the 95 students for the final exam (line 139)?
Knowledge of these methods can help understand any potential bias more and also help replicated the study if needed.

Validity of the findings

No comment.

Additional comments

I praise the authors for conducting this study to look into how DP can be incorporated into PE training in medical school, an important topic that we need more data about. They clearly stated the limitations of the study, highlighting how baise can potentially affect results.

Reviewer 3 ·

Basic reporting

This article had a lot to report and I found the flow a bit hard to follow.
Should re-organize to meet the 3 aims of the paper:
1) determine the reasons why the students cannot master PE skills well in a training course.
2) find the optimal practice times in a PE training course
3) validate the threshold (cut-off times)

Too many tables/figures. Could definitely cut back and focus on the main outcome.

Minor English corrections such as use of "hand-on" on line 85.
Typo in Figure 2 "Dyas" instead of "Days"
Other minor grammatical corrections

Background is fine, but flow could be improved to go along with the aims of the current study. Also, could add some info about PE and what the requirement is for learning.

Experimental design

As above, this article had A LOT to report and I found the flow a bit hard to follow.
Should re-organize to meet the 3 aims of the paper:
1) determine the reasons why the students cannot master PE skills well in a training course.
2) find the optimal practice times in a PE training course
3) validate the threshold (cut-off times)

You also reported on setting that Aim 3 threshold, but don't really mention it in the methods. I think this may be the missing link? First you did a survey on PE skills and what learners and teachers were missing, then you did training for a group of students and looked at practice session numbers and validated the thresholds. Where does the standard fit?

There were methods mixed in with background (and AIMs), and results mixed in with methods. It was hard to figure out what the actual MAIN outcome/goal of this study was since it seems like it all took place simultaneously?

The methods and results need a major re-organization so a reader could follow what is done when and WHY.

The title of the article mentions "MASTERY", however, this is not an example of "mastery-learning" as described in the literature.
1) It was mentioned in the limitations that there was no baseline assessment- this is a key feature of mastery learning
2) all trainees need to reach the "standard", set by experts, which isnt entirely clear in this article in order to achieve "mastery"
3) Need more info in the methods on how the standard is set- the study mentions angoff method. It also seems as though 5 "experts" helped to set the standard- there should technically be at least 8, and how were they selected? who were they? specialties, etc.

None of the other survey findings were addressed? the second highest suggestion from both teachers and learners was enriching the content? Was anything done in that sense?

Validity of the findings

This is hard to assess without the improvement of the other areas. I think the data is there, it just needs to be sorted out and focused. Maybe there are too many Aims and the authors should nail down what's most important.

---

## Round 0.2 · Minor Revisions

Please pay careful attention to the recommendations of Reviewer #1. I do appreciate your time and patience with this iterative process.

Reviewer 1 ·

Basic reporting

1. Language: This version of manuscript has significant improvement in language, however, minor language issues still exist, some of which are concerned with professionial expression in medical education. I would suggest a thorough language check of the manuscript performed with the aid of a (near) native English speaker.

2. Figures: The current presentation of the figures are much easier to comprehend. However, figure legends of the manuscript are not presented in the current version (interestingly, you do have the figure legends in the supplemental materials). A relatively detailed description of the 3 figures should be available; otherwise, it would be quite impossible for the readers to find out the meaning the the axises and other elements in the figures.

For the flowchart (figure 3), I would suggest the authors to keep with the two arms after randomization, instead of combining the two arms during assessment.

Experimental design

Since the authros have got rid of the the questionaire in the main text, the previous Aim 1 was excluded . This makes the article more logical to read.

Do the authors keep record of the time spent in each teaching/learning session for both Cohorts? What was the portion of time spent on learner practicing vs learner receiving feedback? Such result could potentially help you with answering the impact of feedback vs repetion of practice in DP.

Validity of the findings

How to determine whether "the threshold (cut-off cycle) defined by the learning curve" has been validated (Aim 2 in current version)? What are the outcome measures that supports this validation process? The authors are suggested provide further explaination in the Discussion Session to answer this question.

Also, since the authors has mentioned other studies (in Lines 176-179) that was trying to address the repetition components of DP (although not focusing on abominal PE), what are the similarities and differences regarding the findings of your research compared with those previously reported ones? You only mentioned afew in Lines 187-193, but the ones from McGaghie and Bursk et al. are also worth further discussion.

The authors mentioned the the potential bias of more repetition that resulted in more feedback (Lines 213-215), I think this is worth extend your discussion, not just as a limitation, but as an association, or benefit of having repetition of high frequency practice.

Since this is an interesting topic of study, have you considered further research to be performed in the future? Future directions of your reseach, as well as the strength of your study should also be included in your manuscript.

·

Basic reporting

Thank you for giving me the opportunity to review this interesting manuscript that tackles an important undergraduate medical education topic. After revisions, I believe the manuscript now is much clearer in enlisting aims and goals, and language and structure are more fluent and easier to follow. It does help when unnecessary information and data are removed or abbreviated to give the reader a more concise and direct view of the flow of the study and its results, and I believe this what happened here. All the data were appropriately shared and demonstrated in simple figures and tables. It was quite clear why authors wanted to do the study. Finding an optimal practice frequency for one of the important physical exam (PE) skills - abdominal examination - needed for students without prior clinical experience to achieve a pre-defined competence is crucial for their development as competent future physicians. It also assists in developing PE curricula during medical school training.

Experimental design

Deliberate practice (DP), as outlined in the manuscript, was found to be associated with better skills acquisition in medical simulation in general, for example during surgical and procedural training. PE can be looked at as a 'bedside medical procedure' where DP can be applied to reach acceptable level of professionalism. This manuscript directly addresses the question of how many practical sessions, on average, a medical student without clinical experience will need to achieve an agreed upon competence level in performing abdominal examination. Although the number of students involved is relatively small but still the two studies used to answer the question (the learning curve study and the comparative study) were appropriate and help formulating an answer to that question. Applying similar methods with larger number of students, in different learning environment and institutions, with other physical examination domains will be even more informative and educationally useful. Standardizing of assessment was established with excellent k-value and that is very important in this kind of studies.

Validity of the findings

Based on the methods described and the settings in which the study was performed, the findings make sense and reasonable. It will be interesting to know if the number of practices needed to achieve competency will decrease when the ratio of instructors to students is less than the ratio used in the study (between 1:12 and 1:14). Smaller ratio is expected to give more feedback time for the instructors and that might, at least theoretically, enhance the learning curve. Also, doing the final test on a standardized patient with actual findings on abdominal exam (e.g., splenomegaly) would have made the study more pragmatic and powerful.

Additional comments

Overall, I think this is a useful and informative study that can add to body of literature related to physical examination training in undergradulate medical education. I encourage the authors to do similar studies in other aspects of physical exam and consider points mentioned above to make these studies even more robust.

---

## Round 0.3 · accepted · Accept

Thank you for your patience throughout this review process. There are a couple of minor requests associated with the last review that I encourage you to address.

Reviewer 1 ·

Basic reporting

The latest version of the submitted manuscript has improved in language, however, minor points of medical (education) terms should be noted as follows.

1. Line 94, since it is the first time that M.D. appeared, I would prefer using "with/holding a Doctor of Medicine Degree" or similar expressions.
2. Line 136, I think by using "Diagnostic Syllabus", the authors are actually trying to say "Physical Examination Course Syllabus".
3. Line 224, maybe a past participle of "have hypothesized" is better than the current past tense used.

Experimental design

The authors mentioned in the Method part that the Angoff method with 5 judges was usedto determine the competence score, the latter of which is a kind of cut-off score. This helps clarify the approach they used, although 5 judges is the minimum acceptable number of judges involved in a standard setting. Anyway, it's acceptable.

Validity of the findings

The authors have extended their Discussion part in the current version of manuscript. Lines 195-206 has provided improved comparsion of their results with other DP related studies.

Additional comments

Please see to the minor language issue, I think the manuscript is otherwise okay for acceptance.